# A global multi-hazard risk analysis of road and railway infrastructure assets

E.E. Koks[1,2], J. Rozenberg[3], C. Zorn [1], M. Tariverdi[3], M. Vousdoukas[4,5], S.A. Fraser[3], J.W. Hall[1] & S. Hallegatte[3]

Transport infrastructure is exposed to natural hazards all around the world. Here we present the first global estimates of multi-hazard exposure and risk to road and rail infrastructure. Results reveal that ~27% of all global road and railway assets are exposed to at least one hazard and ~7.5% of all assets are exposed to a 1/100 year flood event. Global Expected Annual Damages (EAD) due to direct damage to road and railway assets range from 3.1 to 22 billion US dollars, of which ~73% is caused by surface and river flooding. Global EAD are small relative to global GDP (~0.02%). However, in some countries EAD reach 0.5 to 1% of GDP annually, which is the same order of magnitude as national transport infrastructure budgets. A cost-benefit analysis suggests that increasing flood protection would have positive returns on ~60% of roads exposed to a 1/100 year flood event.

[1] Environmental Change Institute, University of Oxford, Oxford OX1 3QY, UK. [2] Institute for Environmental Studies, Vrije Universiteit Amsterdam, Amsterdam 1081 HV, The Netherlands. [3] World Bank, Washington, DC 20433, USA. [4] European Commission, Joint European Research Centre (JRC), Ispra I-21027, Italy. [5] Department of Marine Sciences, University of the Aegean, Mitilene 41100, Greece. Correspondence and requests for materials should be addressed to E.E.K. (email: elco.koks@vu.nl)

Reliable transport infrastructure is one of the backbones of a prosperous economy, providing access to markets, jobs, and social services[1]. For this reason, the sustainable development goal (SDG) 9 calls for increased access to sustainable transport infrastructure in low and middle income countries. Collectively, new estimates show that low and middle income countries will need to spend between 0.5 and 3.3% of their gross domestic production (GDP) annually (US$157 billion to 1 trillion) in new transport infrastructure by 2030—plus an additional 1–2% of their GDP for maintaining the network—depending on their ambition and their efficiency in service delivery[2,3]. Securing resources for maintenance is crucial to ensure the continued safety and reliability of transport systems, and yet many countries have struggled so far to secure the resources for sustainable maintenance spending[2,3].

At the same time, due to the wide spatial distribution of transport infrastructure, many transport assets are exposed and vulnerable to natural hazards, increasing the costs for national transport agencies and operators. Examples are the major infrastructure damages during Hurricane Maria in Puerto Rico in 2017[4], large-scale road and bridge damages in Sulawesi (Indonesia) during the 2018 earthquake, and widespread infrastructure damage following the Tohoku Earthquake in 2011 in Japan[5]. During the 2015 floods in Tbilisi, Georgia[6], the repair of transport assets contributed approximately 60% of the total damage cost.

The economic and social consequences of natural disasters have attracted more attention recently and the international community, through both the SDG 11 (make cities and human settlements inclusive, safe, and sustainable) and the Sendai Framework for Disaster Risk Reduction, which has called for improved risk management when building and managing infrastructure networks. Further evidence on the damages that infrastructure networks face due to natural hazards at the global level is required to bring useful policy insights and guide possible revisions of infrastructure planning and design standards worldwide. Such results would be particularly important for low income countries where investment needs are the highest, but risk assessments are scarce and disasters impacts on the economy are typically underestimated[7].

However, the scientific evidence is still limited on the global impacts of natural disasters on infrastructure networks, both through direct infrastructure damage and indirect impacts on users and supply chains. Many existing global disaster risk models focus on damaged buildings or affected populations[8–10], with infrastructure exposure being modeled using generalized assumptions on infrastructure density rather than detailed asset mapping. To our knowledge, no global study addresses damaged networked infrastructure at the asset level, such as individual road segments or bridge structures.

Moreover, most studies[8–10] have focused on single hazards in isolation (i.e., flooding), while a full evaluation of risks from natural disasters need to account for the different specific characteristics and geographical extents of natural hazards. Comparisons and quantification of global transportation asset exposure and potential damages under a wider range of hazards is required at the global level to assess the fiscal burden of damage from natural hazards and to quantify the potential benefits of adaptation action.

To fill these gaps, this study presents first estimates of global exposure and risk of road and railway assets to the most frequently recorded and costliest disasters: tropical cyclones (wind speed only), earthquakes, surface flooding, river flooding, and coastal flooding[7]. Other natural climatic and geomorphological hazards (i.e., landslides) are absent from the analysis due to the limited availability of consistent global data. Our study makes use of the latest road and railway asset data available through OpenStreetMap (OSM) and state-of-the-art global hazard maps. As recent studies have shown, OSM can be considered a globally reliable and complete source of transport infrastructure data[11,12] though never previously has this dataset been used for a global risk assessment.

In this study, the global estimates of risk to transport infrastructure are calculated using a conventional risk-modeling framework. We define risk as a function of hazard—the probability and severity of an event with potential to cause harm; exposure—the value of assets subject to the hazard; and vulnerability—the sensitivity of the asset to hazards of given severity[13,14]. Due to limited empirical information and existing research on this topic at a global scale, many assumptions had to be made to conduct this study. For transparency and ease of interpretation, assumptions are clearly set out in both the "Methods" section and in Supplementary Table 1 in the Supplementary Materials. Furthermore, as most assumptions on vulnerability and cost of the infrastructure assets are uncertain, a global sensitivity analysis is performed on the reported infrastructure risk assessments. This results in both a realistic range for the estimated risk and a better understanding of the main factors that influence risk. This assessment allows us to present, for the first time, the annual cost of repairing transport infrastructure damaged by natural hazards (globally and by country), and the direct economic benefits of improving infrastructure standards against flooding. Indirect benefits for users and the economy are left for future research.

We find that ~27% of all global road and railway assets are exposed to at least one hazard. The greatest absolute expected annual damages (EAD) is observed in high income countries, however, relative to GDP, middle income countries are at higher risk. Overall, we estimate that global multi-hazard EAD to transport infrastructure ranges from 3.1 to 22 billion US dollars. On a national level, EAD can reach 0.5–1% of GDP annually. Improving road design by spending about 2% of the road value in better drainage and flood barriers could yield positive return for 60% of the roads that are exposed to at least one flood event. We conclude that it is crucial that countries, when exposed to natural hazards, improve transport planning by systematically including risk information and improving the protection of their most vulnerable and critical assets. Better risk information, obtained from studies like this, will help to avoid spending more on all assets and make it possible to spatially target improvements.

## Results

**Global exposure to natural hazards**. Global exposure of transport infrastructure assets is presented in Fig. 1 across 46,566 regions (Methods). We find that ~27% of the network is exposed to at least one hazard with a 1/250 return period and ~7.5% of the road and railway assets are exposed to a 1/100 years flood event, while in terms of expected annual exposure (EAE, defined as the sum of exposure levels multiplied by their respective return periods), about 0.5% of global assets are exposed to natural hazards. The lowest (relative) EAE is for high-income countries (0.42%) and the highest for lower middle income countries (0.68%). Highest EAE is to surface flooding (Fig. 1d), followed by tropical cyclones (Fig. 1b), river flooding (Fig. 1e), and earthquakes (Fig. 1c). Surface flooding is caused by intense rainfall as a result of local accumulations of water, which can occur in many locations, though the local area and depth of inundation may be small. Tropical cyclones are more extensive in the geographical regions where they occur, while river flooding is constrained to floodplains. The locations of earthquakes correspond to seismically active regions and have greatest impacts in locations where soils are subject to liquefaction. In this study, assets are only considered to be exposed when the probability of occurrence of

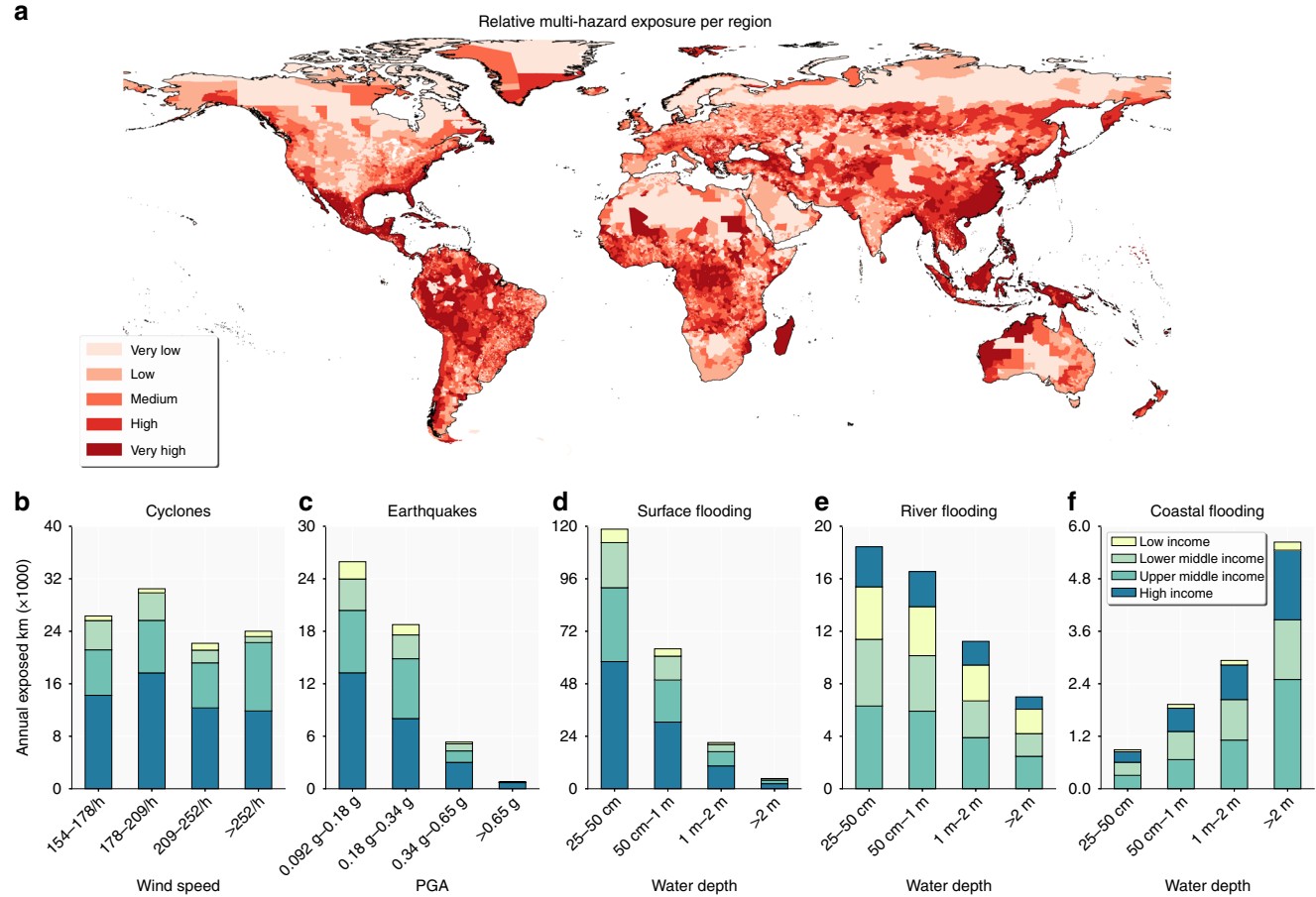

**Fig. 1** Global multi-hazard transport infrastructure exposure. Panel **a** presents the exposure for each region in the world. The classification is based on 20th percentiles. Panels **b**–**f** presents the exposure for the four income groups per hazard and per hazard intensity band. See Methods for further discussions on the justification on these hazard bands

the hazard exceeds the assumed design protection standards of the assets (Supplementary Table 7). For coastal, surface and river flooding specifically, it means that we only assume that infrastructure assets are exposed, and the area inundates, if the severity of the hazard exceeds the design standard.

High income countries have the greatest cumulative length of transport infrastructure (see Supplementary Table 2 for an overview of countries' income group), followed by upper middle income countries and lower income countries. For earthquakes and surface flooding in particular, we see that the amount of exposed kilometers of assets correlates well with the relative shares of each income group in the total amount of infrastructure assets globally (Supplementary Table 5). For river and coastal flooding, however, due to higher flood protection standards[15], High income countries have fewer kilometers exposed. For tropical cyclones and earthquakes, the large share of exposed infrastructure in upper middle income and high income countries is predominantly caused by the geographic occurrence of the hazard. Many of the areas of highest exposure in Fig. 1a align with high cyclone hazard areas: Caribbean, US Gulf and East Coast, Eastern China, South Asia, Japan. This is clearly visible in Fig. 2, where tropical cyclones are the dominant hazard in most of these areas. Earthquake is, for instance, the dominant hazard along the San Andreas Fault and the coastline of Chile and Peru.

A further exploration of Fig. 2 shows that surface and river flooding are the hazards causing the highest expected annual exposure levels in most countries and regions globally (39% and 34%, for, respectively surface and river flooding). Africa is

predominantly exposed to river flooding, which is also translated into the largest share of low income and lower middle income countries' risk in Fig. 1e. Europe and central North America, on the other hand, see a predominant exposure to surface flooding. At a country level, results show that multi-hazard exposure in absolute terms is highest in Japan and China. In relative terms, South Sudan (2.1%) and Madagascar (1.4%) are the countries that experience the highest multi-hazard exposure to their transport infrastructure. These high levels of exposure are primarily driven by fluvial flooding and cyclones for, respectively, South Sudan and Madagascar.

**Vulnerability and risk.** The global EAD to transport infrastructure assets are presented in Fig. 3. These represent direct damages to road and rail assets, and do not include the costs from transport delays and disruption, or wider economic impacts. The total global EAD for all hazards combined ranges from 3.1 to 22 billion US dollars (Fig. 3b), depending on the assumptions made on the vulnerability of roads and construction and repair costs. These values represent between 0.2 and 1.5% of annual global maintenance needs, using the assumptions described in the cost-benefit methods section to assess maintenance needs. The mean EAD for transport infrastructure assets is 14.6 billion USD (1% of annual global maintenance needs).

Approximately, 73% of the global EAD is caused by surface and river flooding (Fig. 3a), followed by coastal floods (15.5%), earthquakes (7.3%), and tropical cyclones (3.8%). Tropical

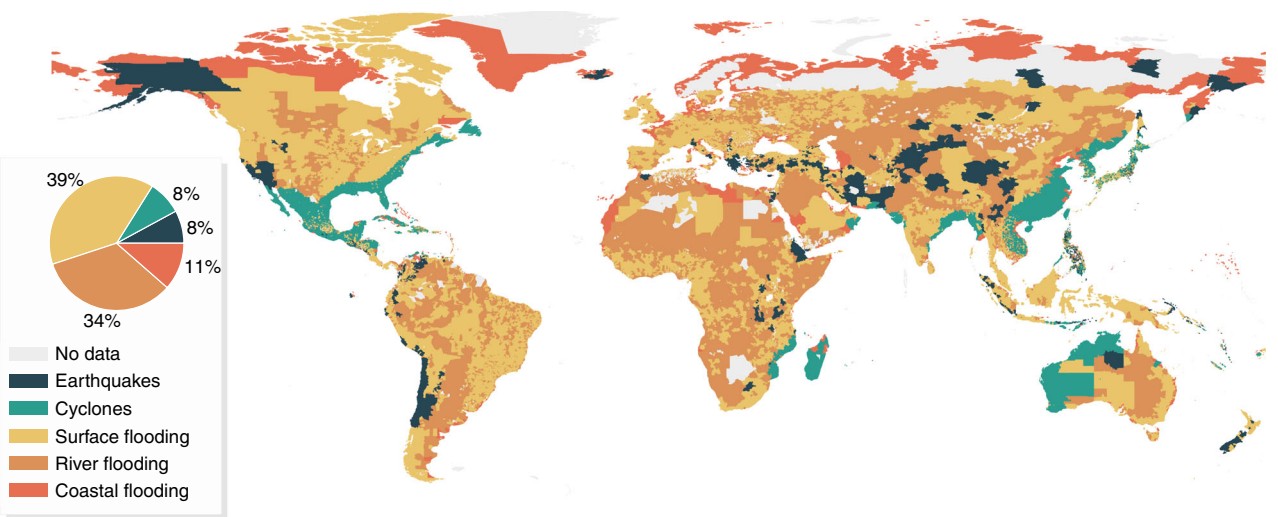

**Fig. 2** Dominant hazard per region. This figure presents the hazard causing the highest transport infrastructure exposure in each region. The pie chart shows the relative percentage of land area (excluding areas with insufficient data) where that specific hazard causes the highest exposure

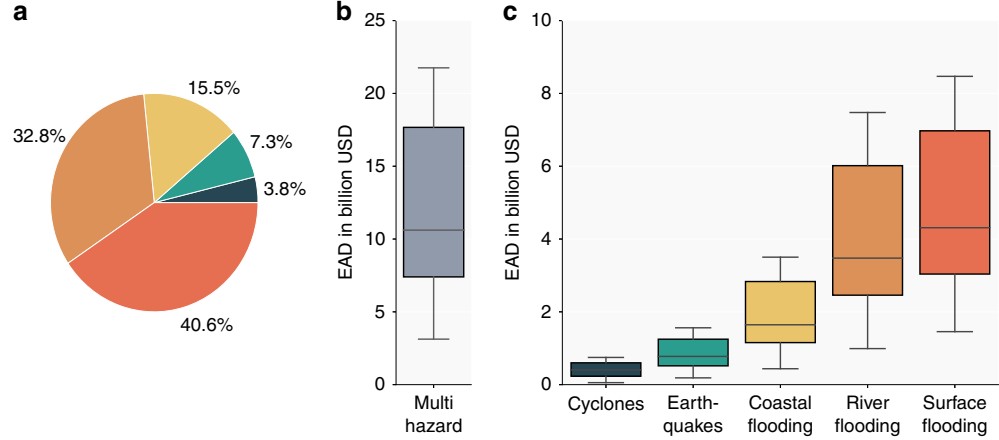

**Fig. 3** Total EAD per hazard. Panel **a** shows the relative distribution of the total EAD to infrastructure assets among the different hazards. Panel **b** shows the calculated range of total multi-hazard global EAD. Panel **c** shows the calculated range of the EAD per hazard. The lower and upper whiskers in panels **b** and **c** represent, respectively, the lowest 25% of the calculated EAD's and the highest 25% of the calculated EAD's

cyclones cause relatively fewer damages compared to their exposure—there are twice as many kilometers of infrastructure exposed to high intensity cyclones than to coastal floods (Fig. 1), and yet cyclone risk is around four times lower. This is because the impact of cyclone winds is largely limited to bridge damage and the cost of removing trees fallen on road carriageways and railway tracks (see Methods and Supplementary Fig. 5). Overall, the EAD in this study are higher compared to the few available estimates on global risk to infrastructure assets. For river flooding, for instance, our EAD values are ~250% higher compared to the global infrastructure risk estimates published in Alfieri et al.[16]. In our view, this is mainly due to the high-resolution representation of infrastructure assets in our study, instead of using a proxy representation as done in Alfieri et al.[16].

Figure 4 presents the results per income group. Intuitively, one would expect exposure of transport infrastructure to natural hazards to increase with income under the assumption that countries accumulate more infrastructure as GDP increases. Indeed, about 50% of global transport infrastructure can be found in high income countries, while only a third is in upper middle income countries and the rest is shared between low and lower middle income countries (Supplementary Table 5). However, high income countries only bare approximately a quarter of the

global risk, while upper middle income countries bare half of the global risk and lower middle income almost a third. This is because as countries move from upper middle income to high income, they invest more in higher protection standards of flood defense[15].

To further explore this and to control for the difference in infrastructure length, we analyze the total EAD per kilometer of infrastructure (Fig. 4). We find a steep increase in total risk per kilometer from low income countries to lower middle income countries, and then a decrease as countries income grows. This bell-shaped curve peaking around the lower middle-income level for total risk per kilometer is largely due to surface and coastal flooding, followed by earthquakes. This can also be observed from Fig. 2, showing that several Central Asian and African countries from this income group have high surface flooding exposure. These results recall the Kuznet curve for environmental degradation[17]. As countries grow richer, they invest in more infrastructure, which increases disaster exposure (and environmental degradation in the case of the initial Kuznet theory)[18]. Absolute disaster damages thus increase as more infrastructure is built. After they reach a higher level of income (in the middle income category), they have enough resources to prioritize higher resilience and they reduce the vulnerability of their infrastructure

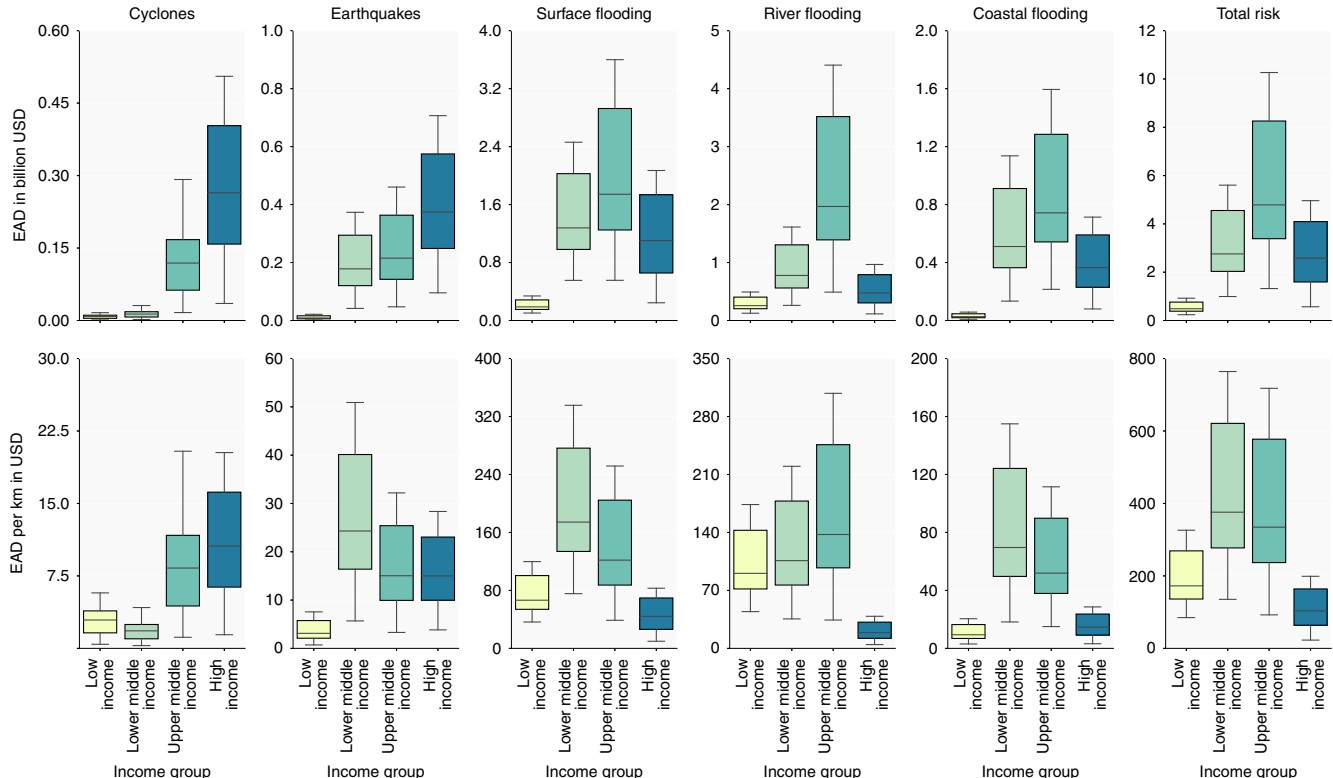

**Fig. 4** Expected annual damages per hazard. The upper row of boxplots presents the absolute EAD for each individual hazard and, in the right-most boxplot, the total multi-hazard EAD. The lower row of boxplots presents the EAD per kilometer of infrastructure for each individual hazard and, in the right-most boxplot, the total multi-hazard EAD per kilometer. The lower and upper whiskers represent, respectively, the lowest 25% of the calculated EAD's and the highest 25% of the calculated EAD's for each income level

assets through investments in more rigorous design standards for transport assets standards and increased flood protection.

The relative distribution of the EAD across transport assets is presented in Supplementary Fig. 5. For most hazards and for most income groups, primary and tertiary roads drive the risk: primary roads as they contribute the highest relative damages, tertiary because they have the greatest cumulative length in all income groups (Supplementary Table 5) and, to a lesser extent, because they are less protected against flooding (Supplementary Table 7).

Road bridges also play an important role in total EAD, for all hazards except surface flooding. For coastal flooding in upper middle income countries, for example, road bridges cause approximately 29% of the EAD, because of a large number of bridges on the coast of China exposed to high coastal inundation levels. In high income countries, over half of the EAD due to tropical cyclones is driven by bridge damages, because, as mentioned before, normal roads and railways only require removal of trees and quick repairs, while a bridge can collapse due to extreme wind levels (see Methods and Supplementary Table 1 for an overview of the assumptions).

The 20 countries in which the transport infrastructure is most affected by natural hazards are presented in Fig. 5 for a range of EAD metrics. When looking at EAD in absolute terms, as shown in Fig. 5a, we find that China and Japan have the highest absolute risk, contributing ~24% and ~11% to global EAD, respectively. In comparison, China comprises ~18% of the global GDP and ~6% of the length of road and rail, whereas Japan comprises around 6% of the global GDP and only 3% of the length. Generally, the countries in this list have either a very high exposure to multiple hazards or high value infrastructure. Isolating these damages per hazard suggests that China's EAD is primarily driven by fluvial

(~40%) and pluvial (~22%) flooding, whereas Japan's EAD is almost fully driven by earthquakes (~67%) and pluvial flooding (~23%). Supplementary Materials provide an overview of these numbers for each country considered in this study.

When looking at EAD as a percentage of GDP (Fig. 5b), the 20 countries with the highest risk are predominately low or middle income countries, consistent with the bell-shaped relationship between damage per kilometer and income apparent in Fig. 4 and discussed in previous paragraphs. For the top five, the median risk is estimated to be between ~0.25 and ~0.42% of the countries' GDP. Interestingly, the EAD as percentage of the total infrastructure value in a country (Fig. 5c) show a somewhat different list of highest affected countries. In particular, the list contains several small islands developing states (SIDS). Results show that the EAD relative to the total infrastructure value is over 100% higher for SIDS compared to the global average, despite the share of primary and secondary roads being comparable (~15% for SIDS vs. ~16% globally). This suggests that these countries have a relatively large amount of high value transport assets exposed to multi-hazards, compared to the global average. This is because in SIDS, most of the available land is exposed to multiple hazards. Finally, EAD in USD/km (Fig. 5d) mainly shows countries which are either heavily exposed, such as Taiwan (China) and Japan, or countries that are small with comparatively few infrastructure assets, such as French Guiana and Fiji.

Several countries appear in three panels in Fig. 5 and can be listed as particularly vulnerable. Myanmar, for instance, is experiencing one of the highest absolute levels of risk to its transport infrastructure, but also the highest risk as a percentage of GDP and one of the highest per kilometer of road. Papua New Guinea shows one of the highest EAD as a percentage of infrastructure value, the second highest per kilometer and is

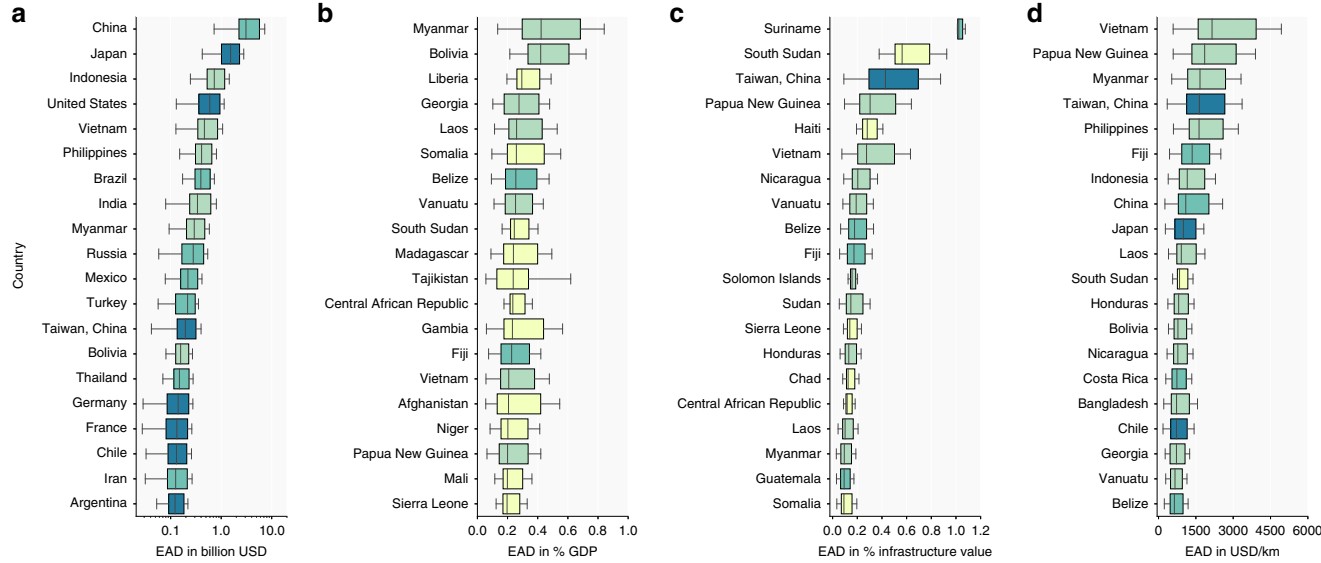

**Fig. 5** Multi-hazard risk per country. Panel **a** presents the 20 countries that have the highest multi-hazard EAD in absolute terms. Panel **b** presents the twenty countries that have the highest multi-hazard EAD relative to the country's GDP. Panel **c** presents the twenty countries that have the highest multi-hazard EAD relative to the country's infrastructure value. Panel **d** presents the twenty countries that have the highest multi-hazard EAD per kilometer of infrastructure in that country. Supplementary Materials provide the underlying data for panels **a**–**d** for all 236 countries

among the countries with the highest EAD as percentage of GDP. For Papua New Guinea, this means that not only its estimated infrastructure value is relatively high compared to its GDP, but also that a relatively high share of its infrastructure is exposed to multiple hazards. Multiple countries experience both high absolute EAD and EAD per kilometer such as Japan, the Philippines, Indonesia, Taiwan (China) and Vietnam. This indicates that for these countries, the infrastructure at risk is predominately of higher value. Overall, the countries in this study which we identify as most vulnerable are consistent with the reported global economic losses and human cost of disasters between 1998 and 2017[7].

**Reducing risk by improving road design.** Figure 6 shows how increasing flood protection standards for roads—by providing barriers or increasing the drainage size—can substantially reduce the global transport infrastructure risk to flooding. Global risk estimates of all floods combined can be reduced up to 42% when upgrading the roads to design standards (Supplementary Table 7) that halves the annual probability of flooding (i.e., upgrading the design standard to withstand a flood with 1/100 return period instead of a flood with 1/50 return period). This would be a 0.1–0.9% reduction in global maintenance costs annually[3] (Methods).

In absolute terms (Fig. 6b), the largest reduction in damages can be achieved by upgrading the design standards for surface flooding, as this type of flood is dominant in all income groups. In relative terms (Fig. 6c), the largest gains can be made for coastal flooding in low income countries, with a reduction of up to 80% in risk. This large gain is primarily due to higher inundation depths for coastal flooding compared to surface and river flooding (Fig. 1e). In spatial terms (Fig. 6a), we observe the largest decrease in risk in Sub-Saharan Africa, South-America, Southeast Asia, and most of Russia and Mongolia. Lowest reductions are mainly observed in North America and South Africa. When comparing Fig. 6a with Fig. 1a, we see that the areas with the largest decreases are also areas that have the highest levels of road exposure. The opposite is true for the areas with the lowest reductions.

Assessing the cost of providing higher flood protection for roads and railways globally is challenging. For new roads, it is estimated that upgrading the drainage system or providing barriers—to increase the design standard or approximately halve the expected damages—costs about 2% more in capital expenditures[19,20]. However, for some existing paved roads, increasing standards would mean rebuilding road sections almost entirely to replace culverts and drains. It thus would not usually make sense to upgrade the standard until the road requires a major rehabilitation. For rural roads, on the other hand, cheaper adjustments can be made to existing roads by digging trenches on the side, also for about 2% of the road cost. As for bridges, foundations can be protected against erosion and scour caused by floods for 1% of the bridge capital value[19,20]. Modern design practice can ensure resistance to wind damage in all but the severest of cyclones.

Despite the difficulties, it is interesting to get a sense of the potential benefit-cost ratio of upgrading roads to reduce the risk of flooding to road assets. To do so, we perform a cost-benefit analysis on each road segment (CBA, Methods). The CBA estimates the benefit-cost ratio (BCR) of upgrading the road by spending 2% of the road's value on barriers and better drainage. For roads that are not exposed to any hazard, such an investment does not have any benefit and thus has negative returns. For roads that are exposed to floods, we assume that this 2% cost increase allows to multiply the standard of the roads expressed in return period by two (i.e., the road can withstand a flood with 1/100 return period instead of a flood with 1/50 return period).

We find that such an improvement only has a BCR higher than 1 for 4.5% of all kilometers of roads. This is not surprising given that only 7.5% of all roads are exposed to at least one flood event with a 1/100 year return period. Zoomed in on the different income groups and road types, the highest share of kilometer of roads with a BCR higher than 1 is around 10%, which is for upgrading secondary roads in lower middle income countries. For several income and road type categories, the share of kilometers of roads with a BCR higher than 1 is below 1% or even zero (Supplementary Fig. 7).

However, results show a BCR higher than 1 for around 60% of all kilometers of exposed roads (to at least one flood event with a

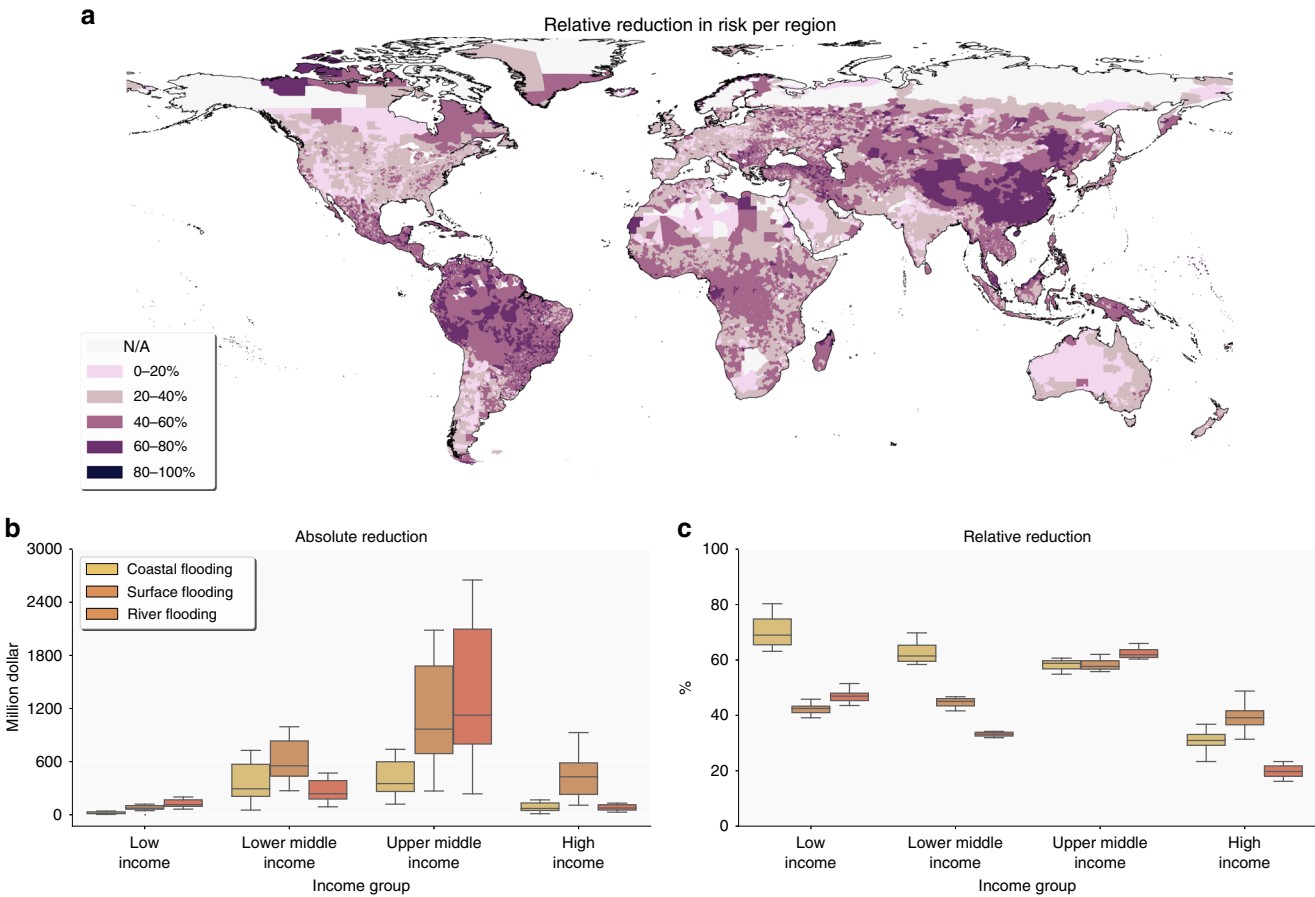

**Fig. 6** Risk reduction due to flood design-standard upgrades. Panel **a** presents the total combined reduction in EAD per region. Panel **b** presents the absolute reduction in EAD for coastal, surface and river flooding. Panel **c** presents the relative reduction in EAD for coastal, surface, and river flooding

1/100 year return period). Improving design standards of exposed primary and secondary roads in upper middle income countries to better cope with surface flooding is beneficial for ~85% of these roads, with an average BCR of ~6. Important to emphasize is that in this study, we only focus on the direct asset damages. When including network disruptions and the wider economic impacts, total avoided losses are expected to increase, making investments in adaptation potentially more beneficial in more places. These results highlight the value of having hazard information for designing roads, which makes it possible to target improvements on exposed roads only. In the absence of any hazard information, spending 2% more on every road would be very cost inefficient.

**Model sensitivity**. An inherent problem with global studies, and for disaster risk modeling on this scale in particular, is the large number of assumptions required to make in such a data-scarce analysis. This is not only the case for the assumptions taken in the analysis of this study (Supplementary Table 1), but also for the approaches taken in the development of the external input data that is being used. As addressed by Ward et al.[21], it is essential to quantify and understand the model uncertainties when applying such global models for disaster risk management. For flood risk analysis in particular, a large part of the uncertainty is on the hazard side. A recent study on uncertainty in coastal flood risk assessment shows that a change in resolution from 10 to 100 m of the digital elevation map could change the estimated EAD by 200%. Moreover, existing errors in flood defense height datasets can change the risk up to 60%[22].

As we do not create any hazard maps in this study, we focus on the quantification of the uncertainties in the loss estimation. As

shown in De Moel and Aerts[23], large uncertainties arise in maximum loss values and the shape of the fragility curves. According to De Moel et al.[24], the shape of the damage curve accounts for up to 45% of the total sensitivity in loss modeling outcomes, and the value of the elements at risk accounts for up to 10% of the total model sensitivity. Supplementary Fig. 7 presents the outcome of a global sensitivity analysis of the risk analysis in this study. For road and railway assets, we find similar results across the hazards. Across earthquakes and floods (coastal, surface, and river), road carriageway damages are particularly sensitive to the choice of fragility curve and the assumed repair costs.

Variation in fragility models contribute the most to the uncertainty in the loss estimates, with around 60% of damage explained through this assumption and with the variation in reconstruction costs responsible for 20% of the total loss. For road and railway bridges, on the other hand, the reconstruction cost is found to be the most dominant driver of the damage estimations, constituting up to ~60% of the losses for cyclones and earthquakes, and up to ~40% of the flood damages. Reducing this uncertainty is particularly challenging as it would require location-specific damage curves and repair costs, which depend on local geographic and economic circumstances. For example, repair costs depend on the efficiency of the local transport authorities and the local cost of raw materials. Despite the difficulties, these geographically varying fragility curves should be developed in the future to reduce uncertainty and improve the damage estimates.

## Discussion
This is the first study to have quantified the global risk to transport infrastructure assets for multiple natural hazards. We have used

state-of-the art global hazard mapping, combined with innovative analysis of approximately 50 million km of transport network data included in OSM, and assumptions about the fragility of transport infrastructure derived from a variety of sources. The study demonstrates the potential for conducting infrastructure risk analysis at a high spatial resolution on a global scale.

The total global EAD for all hazards combined ranges from 3.1 to 22 billion US dollars, with a mean EAD of 14.6 billion USD. Approximately, 73% of the global EAD is caused by surface and river flooding, followed by coastal floods (15.5%), earthquakes (7.3%), and tropical cyclones (3.8%). Sensitivity analysis has revealed the importance of understanding asset fragility. The results for overall transport infrastructure exposure and risk are broadly in line with previous risk analyses of natural hazards (e.g., IPCC SREX[25]), which demonstrate greatest absolute levels of risk in high income countries, but higher risk as a percentage of GDP in middle income countries.

At the global level, EAD are small compared to the budget required for maintaining reliable transport networks (0.2–1.5%). One might thus conclude that building more resilience is further down the list of priorities, after ensuring sustainable sources of funding for regular maintenance. However, our results reveal geographical disparities in exposure and risk, with for example the particular vulnerability of transport infrastructure in small island developing states. Countries like Fiji already spend 30% of their government budget every year in maintaining their transport network[26], and the bill becomes prohibitive when damages from natural hazards are added on top. In other words, we find that for several countries and regions, investing in transport asset resilience should be a priority.

We have found that improving road design by spending about 2% of the road value in better drainage and flood barriers could yield positive return for 60% of the roads that are exposed to at least one flood event and over 80% of the primary and secondary roads flooded on average every year in upper middle income and tertiary roads in lower and middle income countries. Of course, care should be taken with the interpretation of these results, as local road conditions are unknown in this study and a generalized approach is taken. Nonetheless, it is clear that there are significant benefits to be gained from improving the resilience of exposed transport infrastructure. These are expected to be low-regrets investments in the context of a changing climate[8,9,27]. Multiple studies indicate upwards trends in flood risk, which we would also expect for transport infrastructure, as flooding constitutes a large share of the total EAD (all flood hazards constitute 89% of the risk).

We conclude that it is crucial that countries, when exposed to natural hazards, improve transport planning by systematically including risk information and improving the protection of their most vulnerable and critical assets. There is a need for better risk information to avoid spending more on all assets, but being able to spatially target improvements. The economic and social benefits to be gained from doing so would go well beyond direct infrastructure damage. Indeed, studies[28–31] that estimate the economic impact of disasters through transport-economic models that account for the impact of transport interruption on the ability of supply chains to maintain production, conclude that indirect losses as a result of infrastructure failure represent a large share of the total cost of disasters. Further, the additional cost of building in more resilience (about 2%) can easily be offset by better planning and higher efficiency in spending and service delivery, which can halve total spending needs[3].

## Methods

**General approach.** An overview of the approach taken is presented in Supplementary Fig. 1. Due to the large size of all data sources (both in storage and in information), we have split the analysis over 46,566 regions based on the GADM

administrative level 1 and 2 datasets[32]. By using parallel and cloud computing, runtimes have been brought down to a reasonable time-scale, allowing for a global risk analysis with this level of detail. The remainder of this section will explain the analysis in depth.

**Global hazard data.** This study includes earthquakes, tropical cyclones, and surface, river and coastal flooding. For the exposure analysis, hazard data is reclassified into hazard intensity bands, as shown in Supplementary Table 3. The earthquake bands are based on USGS ShakeMap intensity mapping to peak ground accelerations (PGA). For tropical cyclones, we take a similar approach using the Saffir–Simpson scale[33]. There are no widely recognized intensity bands available for floods, so these bands are based on empirical evidence of loss intensity.

In recent major earthquakes (e.g., Loma Prieta 1989, Kobe 1995, Canterbury Earthquake Sequence 2010–2011, Sulawesi 2018), roads and railway tracks have been widely damaged through soil liquefaction leading to irregular settlement of surfaces, lateral spreading toward waterways, and the uplift of buried services. With direct shaking damage to road and rail expected to be minimal, we adopt liquefaction susceptibility as a proxy for potential road and rail damage across the different studied return periods. Damages can range from superficial with minimal clean-up costs, to complete replacement[34,35]. The likelihood of surface rupture damage to assets in close proximities to fault lines and permanent ground displacements are not considered herein.

As state-of-practice in situ testing for assessing liquefaction potential is not feasible at the global scale, we adopt the geospatial prediction models of Zhu et al.[36] to create a global liquefaction susceptibility map[37]. The models relate common ground-motion intensity measures with geospatial parameters relevant to liquefaction susceptibility. Calibrated to 27 earthquake events, the models have since shown promising predictive capacity at high resolutions[38]. For our global susceptibility model we combine the inland and coastal models of Zhu et al., with the coastal model applied within 20 km of a coastline. Liquefaction susceptibility is computed at a 1.2 km grid resolution based on a global $V_{S30}$ (30 m averaged shear-wave velocity) dataset[39]. Other required datasets are collated for: rivers[40–42], ground water[43], precipitation[44], and land mass[45]. Susceptibility is grouped into five classes: very low, low, moderate, high, and very high[34,36]. Very low liquefaction susceptibility is assumed where $V_{S30} > 620$ m/s. The compiled dataset is available in the supplementary material (Supplementary Fig. 4).

Liquefaction susceptibility is combined with earthquake ground shaking hazard as a trigger for liquefaction; the damage ratio is greatest in areas with high liquefaction susceptibility combined with high ground shaking intensity. Ground shaking hazard is represented by the global earthquake hazard maps produced for the UNISDR Global Assessment Report 2015[46]. These maps present the expected severity of ground shaking as PGA (in cm/s$^2$), for five return periods between 1/250 and 1/2475 years. The hazard maps are an output of probabilistic seismic hazard analysis with global coverage. The coarse resolution of the analysis allows for global coverage, but a trade-off is that it limits the consideration of local or unknown faults not previously captured in historical catalogs, therefore they could underestimate hazard locally in some areas.

Tropical cyclone hazard is represented by global cyclone hazard maps generated for the UNISDR Global Assessment Report 2015[46]. These maps provide the distribution of cyclone wind speed (peak wind speed of 3-s gusts, in km/h) for five return periods between 1/50 and 1/1000 years. The maps are an output of probabilistic cyclone analysis based on perturbation of historical cyclone tracks and wind-field modeling. The data provides coverage of the Northeast Pacific, Northwest Pacific, South Pacific, North Indian, South Indian, and North Atlantic basins. The data does not include the effects of extratropical cyclones or convective storms in these basins or other areas.

River (caused by rivers overtopping their banks) and surface (caused by extreme local rainfall) flood hazards are represented by the Fathom Global pluvial and fluvial flood hazard dataset[47]. This is a 3-arcsecond (c. 90 m) resolution gridded dataset showing the distribution of maximum expected water depth in meters. The hazard maps are provided for ten return periods (1/5–1/1000) with global coverage between 56°S and 60°N. We apply the undefended flood hazard maps, which do not consider the effects of flood protection on inundation. Instead, the flood design standards for road and rail are implemented from the FLOPROS database[15], as discussed below. As reported by Sampson et al.[47], the global model shows reasonable agreement with local flood hazard data, particularly for large rivers, though for smaller rivers there is potential that this data underestimates flood hazard, typically capturing between two-thirds and three-quarters of flooded area shown in local data.

Coastal inundation maps are generated using the hydrological model LISFLOOD-FP[48]. Topographic information at 3″ horizontal resolution is available from the MERIT-DEM[49]. Inundation simulations take place at 90 m resolution, along coastal segments that include 75 km of coastline each and land up to 100 km from the coast. Neighboring segments are overlapping approximately 25 km of coastline to avoid gaps in the resulting inundation maps. Further details about the implementation of the inundation modeling can be found in Vousdoukas et al.[50].

Flood simulations are forced by extreme sea levels (ESLs) obtained from wave and storm surge reanalysis, combined with tidal information[27]. Waves are simulated using the WAVEWATCH-III model[51], and storm surges using the

DFLOW-FM model[52]. Wave contributions to ESLs are considered equivalent to the wave setup, expressed as 20% of the significant wave height. Following, nonstationary extreme value analysis is applied to estimate ESLs at seven different return periods (1/10–1/1000)[53]. Flood simulations are forced by ESLs obtained from reanalysis of waves and storm surges, combined with tidal information[27], including also the effect of tropical cyclone storm surges.

**Global infrastructure data**. All road and railway data are based on open access data from OSM. Supplementary Table 5 provides summary statistics of the road and rail network used in this study. Recent studies have shown that the accuracy of OSM has increased substantially over the last few years[11,12]. Meijer et al.[12] shows that the most comprehensive open global road datasets range from 7 to 32 million kilometers of roads. The total length of road infrastructure assets extracted from OSM is over 63 million kilometers and according to Barrington-Leigh et al.[11], OSM was over 80% complete in January 2016, with missing data most likely in lower tier roads. The other main caveat of OSM data is the correct classification of the infrastructure[54], which is accounted for by aggregating the roads to only four classes (primary, secondary, tertiary and other roads; Supplementary Table 4 shows the mapping of the OSM classification to these five classes). Primary roads can be described as all major highways and trunk roads. Secondary roads as all major provincial and subnational roads. Tertiary roads are considered to be important local roads, often linking secondary or primary roads with each other. For this study, we only consider a subset of all roads, namely those classified as primary, secondary, or tertiary roads (Supplementary Table 4). However, the total length of included carriageways still amount to ~50 million kilometers. All roads classified as other are not included. Bridges considered in this analysis are those road and railway assets that are tagged as bridges in the OSM database. Due to a lack of a global bridge database, we believe this is the most complete and readily available open dataset.

**Estimation of damage and risk**. The conditional probability of failure of an infrastructure asset when subject to an extreme hazard depends upon the design of the asset and a variety of site-specific conditions, which are difficult to incorporate in a global assessment. However generic understanding of the fragility of infrastructure assets with respect to some natural hazards does exist (as documented below) and where it does not exist we have made well-informed and transparent assumptions which were then subject to a sensitivity analysis (Supplementary Table 1).

Infrastructure damages are estimated using a variety of sources of replacement cost data and fragility curves. For the uncertainty and sensitivity analysis, we make use of the SALib Python library[55], allowing us to identify the relative importance of each parameter in the loss estimation. The range of parameters tested for each hazard and asset combination is given in Supplementary Table 1. Estimation of road development costs and maximum replacement costs are based on the road costs knowledge system (ROCKS), an empirical study of road-related projects[56]. The ROCKS database provides construction cost per kilometer for different infrastructure types, based on historic infrastructure projects. As the database does not include statistics for all countries and all road types, we use the average cost of construction for *paved 4 lanes*, *paved 2 lanes* and *gravel roads* for different World Bank regions (Supplementary Table 8). For each unique asset, we calculate the damages between 40 and 60 times for each hazard and for each return period, depending on the number of parameters that are varied (see Supplementary Table 1). For each asset type, a set of parameter values is determined using a Morris sampling approach[57], allowing for an optimal distribution of parameter values between the bounds. The results of the sensitivity analysis show how changes in each parameter that is included influences the estimate risk value.

Using this set of unique parameter values, damage states are estimated by relating the intensity of the hazard to a damage probability using predetermined fragility functions, such as depth-damage curves used in flood risk literature. For each asset in each hazard scenario, this damage probability is multiplied by our assumed reconstruction cost for the respective asset (Supplementary Table 8). When knowing the damage to this asset for each hazard event, the risk is calculated by using a trapezoid function to estimate the area under the exceedance probability loss curve[8,23,24].

To estimate the infrastructure value (as used in Fig. 5), we used a similar approach as estimating the possible damage to a road. For the infrastructure value, we would also need to know the type, the width and, in the case of a road, whether it's paved or unpaved. As such, we again take the set of parameters we use for the sensitivity analysis and use this to calculate a range of possible infrastructure values for each infrastructure asset. To get one best guess value, we use the median of these outcomes.

Seismic fragility curves generally relate ground shaking intensity to a damage ratio. As established earlier, damage to road and rail infrastructure is generally due to liquefaction or ground displacement. Due to the lack of local data to inform the potential for ground displacement, we combine the global liquefaction map developed for this study with global PGA to estimate damage due to earthquakes. Because of the limited availability of globally appropriate fragility curves concerning the relation between PGA, soil liquefaction, and road and railway damages, we use a fragility matrix, assuming a different level of damage to the

infrastructure, based on the combination of PGA and liquefaction susceptibility (Supplementary Table 6).

We have found very little empirical information about direct cyclone wind damage to road and railway assets. As such, we assume that extreme cyclone winds mainly result in the clean-up cost of trees and minor reparations. We assume that trees will fall if the cyclone winds exceed 42 m/s (151 km/h), as presented in Virot et al.[58]. To estimate whether a road carriageway or railway track may be affected by tree fall, we make use of a global tree density map, developed by Crowther et al.[59]. We consider a 100% probability of a tree falling on the infrastructure asset if the tree density around the asset is at least 10/km$^2$.

For flooding, the fragility curves for road carriageways and railway tracks are shown in Supplementary Fig. 2. The curves are based on the studies of Espinet et al.[31] for paved roads and for unpaved roads. Flood curves for infrastructure tend to be linear in shape, mainly due to the limited empirical information available to improve them. Unfortunately, for most of the roads extracted from OSM, we do not know whether they are paved or unpaved. By using the kilometers of paved and unpaved roads from the CIA World Factbook, we were able to get the percentage of paved roads for each country. This, however, only gave us a total percentage based on all roads in the country. To get a percentage for each road type in a country, we compared the share of each road type with this total percentage of paved roads. Let us explain this through an example. Let us say the total percentage of paved roads in a country is 80%. We first compare this number with the percentage of primary roads in the country. If the percentage of primary roads in a country is 30%, we assume that all the primary roads are paved. This means we have 50% of paved roads left. Now we compare this with the secondary roads. Let us say 30% of the roads in the country are secondary roads. Following the same reasoning, this means all the secondary roads are paved and we have 20% of paved roads left. The share of tertiary roads in this country is 40%. As we only have 20% of paved roads left, we assume that half of the tertiary roads are paved and the other half is unpaved. All other roads in the country are considered to be unpaved. For many countries, it is unknown what the percentage of electrified railway is in that country. As such, the percentage electrified vs non-electrified is fully incorporated into the sensitivity analysis.

Flood design standards for road carriageways and railway tracks are shown in Supplementary Table 7. Due to the limited availability of studies, we keep the design standards constant among countries within each income group. We assume that protection design standards differ between income levels, not because of design but because of deterioration over time. We assume that countries with higher GDP levels have more funding available for periodic and routine maintenance of their assets, therefore keeping the design protection standards up to the initial level.

For road and railway bridges, the relation between PGA and damage ratio has been more widely studied. Hence, we are able to make use of already developed fragility curves[60]. The curves are presented in Supplementary Fig. 3. For cyclone and flooding, we assume that bridges are designed to withstand hazards at half the exceedance probability of road carriageways or railway tracks (Supplementary Table 5). For example, if a road carriageway is designed to a 1/100-year event, a bridge structure is designed for a 1/200-year event for the same road type (i.e., primary). We assume that bridges are built up to a certain design standard and a certain hazard threshold (i.e., wind speeds of >250 km/h or inundation levels of >5 m). If both the hazard threshold and the design standard are exceeded, the bridge is assumed to collapse, resulting in 100% loss.

**Cost-benefit analysis and global maintenance costs**. As acknowledged in the main text, estimating the cost of adaptation, and the corresponding net benefits, is difficult. As we think that it is still interesting to get a sense about the potential benefits of adaptation, we performed a cost-benefit analysis on each road segment. The CBA estimates the BCR of upgrading the each road by spending 2% of the road's value on barriers and better drainage (assuming roads are upgraded at the end of their lifetime). High level estimates suggest that these interventions would be sufficient halve the annual probability of flooding[19,20]. For roads that are not exposed to flooding, this investment of 2% of the road value would of course be at lost. We like to emphasize that this is an ad hoc approach, purely for illustrative purposes of what the net benefits could look like.

The benefits are assumed to be the net present value (NPV) of the avoided losses as a result of upgrading the road to protection standards that halves the annual probability of flooding (i.e., upgrading the design standard to withstand a flood with 1/100 return period instead of a flood with 1/50 return period). The costs are estimated based on the NPV of the road upgrade cost to the higher design standard, periodic maintenance and routine maintenance, and are kept equal for all countries and income groups. The upgrade costs are assumed to be 2% of the road value[19], periodic maintenance is assumed to be occurring yearly with a cost of 0.075% of the road value, and the routine maintenance is assumed to happen every 6 years with a cost of 5% of the road value. These assumptions were taken from several calibrations of the HDM4 model[61] on various countries in which the World Bank recently invested in road projects. Primary roads are assumed to last 20 years, secondary roads 15 years and tertiary roads 6 years. The main assumption that differs between the income groups is the discount rate, which we set at 12% for Low Income countries, 9% for lower middle income countries, 6% for upper middle income countries and 3% for high income countries. The difference in discount

rates can be interpreted as the different cost of capital in each of the different income groups and reflect higher risks in low income countries. They are also representative of the discount rates used for public investments in these country groups. The sum of the estimated periodic and routine maintenance over all road segments is used in the paper as the total global maintenance costs.

## Data availability

All transport data is based on OpenStreetMap (OSM), which can be freely downloaded. The planet file used in this study is downloaded at July 17, 2018. Global earthquake and cyclone hazard data is available from UNISDR Global Assessment Report 2015 data portal (https://risk.preventionweb.net). Global fluvial and surface flood hazard data (May 2017 version) is used with the permission of Fathom Global. The coastal flood maps are developed by the Joint Research Centre of the European Commission. The global liquefaction map is freely available to download (https://doi.org/10.5281/zenodo.2583745). Other data sources are provided in the main text and Methods. The source data underlying Figs. 1–6 and Supplementary Figs. 5 and 7 are provided as a Source Data file.

## Code availability

All source code is available through https://github.com/ElcoK/gmtra. All results and figures can be reproduced through the source code.

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

## Acknowledgements

E.K., C.Z., and J.H. were supported by the UK Engineering and Physical Science Research Council under grant EP/N017064/1: MISTRAL: Multi-scale InfraSTRucture systems AnaLytics.

## Author contributions

E.K., J.R., M.T., J.H., and S.H. designed the study. S.F. provided the access to geospatial hazard data. M.V. developed the coastal flood maps. C.Z. developed the global liquefaction map. E.K. performed all the calculations. J.R., M.T., and S.H. collected the underlying information for the vulnerability analysis. All authors contributed to the paper and gave final approval for publication.

## Additional information

**Competing interests:** The authors declare no competing interests.

