## [Peer Review File · Nature Communications]

Reviewers' comments:

Reviewer #1 (Remarks to the Author):

This paper attempts to evaluate multi-hazard natural disaster risks of road and railway infrastructure assets on a global scale, and conducts international comparisons. It is a great masterpiece that has accomplished a magnificent attempt with methodology & data that can be compared as much as possible. Only from this point, this article is valuable enough. I would like to show my best respect as one who has studied related topics.

This paper is of article, not of academic paper, therefore it may not be necessary for the specific risk calculation procedure details to be written. However, since various figures (calculated risks) have appeared and discussed, I would be happy if you could write with consideration of your readers more trusted. In particular, it is not clear how "Loss calculation" in Figure S1 is done. The authors may indicate the minimum calculation formulas so that readers can understand what kind of calculations are done, or arrange better the relevant references instead.

Reviewer #2 (Remarks to the Author):

The authors attempt to define the multi-hazard risks for road and railway infrastructure assets around the globe, using global models and bulk dataset. Overall, the manuscript is sufficiently sophisticated, both the introduction and conclusions are clear, and figures and tables are very informative. Still, I am curious about the applicability of the results of this study although the methods and their level of uncertainty are well documented. My comments are as below.

1. The global infrastructure risk estimated in this study is compared with precedential studies, as mentioned in L.129-133. As the local (national level) infrastructure risks are also quantified, could you show a comparison of some of these with previous studies which also analyze local risk?
2. If my understanding is correct, the periodic maintenance and routine maintenance costs, maintenance period for each road, and the discount rate, are assumed without any relative references, which may present an additional uncertainty to the analysis. I think these assumptions also require sufficient reasoning, as is given for the other assumptions.

3. Perhaps the impact of earthquakes is limited to soil liquefaction due to data limitations. I wonder if it would be possible, at least to consider the cost of removing trees fallen onto roads and railway tracks which were caused by earthquakes, by combining a global tree density map with the dataset related to earthquakes, along with that of cyclones. Otherwise, it should be clarified in L.56 that this study focuses solely on the risk of soil liquefaction associated with earthquakes to avoid any misunderstanding.

4. I would like the authors to provide further discussion regarding the relevance of the results with regard to current international policy schemes for alleviating the damage caused by natural disasters (e.g. SDGs 11.5, Sendai Framework for Disaster Risk Reduction 2015-2030 etc.) to reinforce the policy implications of this study. How could the expected outcomes be useful for policy makers?

In addition, as minor comments,

1. L.257, 263: BCA -> BCR? Otherwise, please define it.
2. L.282, 283: 45 per cent and 10 per cent may be uniformly written as 45% and 10%.
3. L.283: Fig S7 -> Fig S6?
4. L.309: What is SREX? Please explain it.
5. L. 323: “~a changing climate.” some references should be added.
6. L.409: “Flood simulations are forced~(1/10 to 1/1000)” is redundant in the same paragraph.
7. It would be helpful for the reader to add a description for line and error bars in the box-and-whisker diagrams in Fig 3 B and C in the caption or figure.
8. Although I looked at Table S4, I am not sure of the difference between primary, secondary, and tertiary roads. Please clarify this within the main text.
9. For the caption of Table S7, I’m wondering if the table represents the design standards for not just pluvial flooding, but all flooding taken into account in this study.
10. I think a world map representing which income class the regions belong to would be intuitive to complement Figs 1, 2, and 6.
11. Unfortunately, I could not understand how to estimate the sensitivity of parameters shown in Fig S6 even though I read the methodology section. In addition, “4L vs 2L” and “2L vs 1L” should be defined although I could surmise that they were related to the number of road lanes.

Reviewer #3 (Remarks to the Author):

The manuscript presents a novel global analysis of the risks from multiple hazards affecting road and railway infrastructure, using recent and detailed global datasets and explores cost effective strategies for reducing risk. The overall analysis and outcomes, also on cost aspects and regional focus, are a useful and timely contribution that can be of interest to academics and practitioners across various disciplines. The authors are to be applauded for their work on the processing of the large and complex datasets required for the analysis. The methods are well described, including various aspects concerning the uncertainties, sensitivities and assumptions. The paper is clearly written. I believe the paper could be published after addressing the following points:

1. General observation: After reading the manuscript, the reader has a good impression of the research, the assumptions and sensitivities in the used data, applied analysis and the results. However, after the first two paragraphs of the introduction, the reader is left in doubt about the importance of this research: is it mainly a minor (financial) issue for only a few countries or should the outcomes be considered a global alarming message with respect to foreseen future trends (more people, more infra, more traffic, more flooding, more extreme weather events?) and hence could improve planning, even though roads are not maintained globally, but tend to be managed by different levels of government within countries. So besides considering the research a starting point and aim for further research on the economic consequences, the authors could guide the reader by elaborating a bit more on the relevance of the results.

2. Line 29-32: To add to the relevancy of the work, I assume there should be more references available that relate to future infrastructure patterns and investment estimation (beyond 2025?), besides the preliminary paper included now?

3. Line 57-61 / 416-429: The asset dataset is a crucial element in the analysis. The authors mention their source of infrastructure data is Openstreetmap, which, as far as I know, is a crowdsourced mapping initiative. Even though this dataset appears quite extensive, the following issues could be explained a bit more:

a. Do the authors know if there is a geographical bias in road coverage of the OSM dataset, potentially in relation to the crowdsourced nature of data gathering? Could there be consequences for the outcomes? The manuscript does not mention this. From Table S5 it appears that 40% of the roads used in the analysis are tertiary roads in high income countries (covering over 50% of all the roads used). Are the authors confident the road coverage in the other lower income groups (where hazard vulnerability is higher) is of the similar level/quality?

b. It would help the reader to assess the quality of the dataset if table S4 would include the km covered per category

c. The manuscript does not enclose any details on how the so-called planet-file was processed.

4. Line 146-147: does the EAD per km also take into account differences between urban and rural roads, in the sense of traffic affected?

5. Lines 174-174, Figure S1 and Table S1: According to Figure S1 and Table S1 the loss calculation and the assumptions are distinguishing between paved and unpaved roads. Can the authors explain how they applied this categorization to the OSM data? Same goes for the railway categorization mentioned in Table S1.

6. Line 167 and Table S7: are tertiary roads considered part of "other"? This seems confusing with the "other roads" category in Table S5.

7. Line 77 and 337: Besides a technical reason, the authors do not further explain why they have done the analysis at the level of the GADM regions and not, for instance, report the outcomes on a raster with equal cell sizes. How was this aggregation done, this is not explained? The regions vary in size, shape and characteristics, which makes the maps in Figures 1, 2 and 6 a bit challenging to interpret, while most results are also presented on a country or WB region level.

8. Figure 5: How is the value of the country infrastructure, as displayed in Figure 5c, determined? I could not find a description for that.

Minor remarks:

9. Lines 197-198: duplication of text

10. Line 164: I think this points to the wrong figure?

11. Line 257 and 263: BCA? Do you mean BCR?

12. Line 283: I think this points to the wrong figure?

13. Line 301: 60 mln km? I thought "other roads" were not included (line 426-427), so shouldn't this say ~50 mln km.

14. Lines 405-409 / 409-415: duplication of text

Overall response to reviewers

Reviewer #1 (Remarks to the Author):

This paper attempts to evaluate multi-hazard natural disaster risks of road and railway infrastructure assets on a global scale, and conducts international comparisons. It is a great masterpiece that has accomplished a magnificent attempt with methodology & data that can be compared as much as possible. Only from this point, this article is valuable enough. I would like to show my best respect as one who has studied related topics.

Thank you for your kind words!

This paper is of article, not of academic paper, therefore it may not be necessary for the specific risk calculation procedure details to be written. However, since various figures (calculated risks) have appeared and discussed, I would be happy if you could write with consideration of your readers more trusted. In particular, it is not clear how "Loss calculation" in Figure S1 is done. The authors may indicate the minimum calculation formulas so that readers can understand what kind of calculations are done, or arrange better the relevant references instead.

All the code is made available through GitHub <https://github.com/ElcoK/gmtra>, including in-line documentation. Additionally, a ReadTheDocs (<https://gmtra.readthedocs.io>) is available with further documentation on the workflow and how to get things up and running. The approach to calculate the damage and risk is available in the methods section, under "Estimation of fragility, damage and risk". We added the following lines to further clarify the damage calculation approach we took:

Using this set of unique parameter values, damage states are estimated by relating the intensity of the hazard to a damage probability using pre-determined fragility functions, such as depth-damage curves used in flood risk literature. For each asset in each hazard scenario, this damage probability is multiplied by our assumed reconstruction cost for the respective asset (**Table S8**). When knowing the damage to this asset for each hazard event, the risk is calculated by using a trapezoid function to estimate the area under the exceedance probability loss curve.

Reviewer #2 (Remarks to the Author):

The authors attempt to define the multi-hazard risks for road and railway infrastructure assets around the globe, using global models and bulk dataset. Overall, the manuscript is sufficiently sophisticated, both the introduction and conclusions are clear, and figures and tables are very informative. Still, I am curious about the applicability of the results of this study although the methods and their level of uncertainty are well documented. My comments are as below.

Thank you for your kind words!

1. The global infrastructure risk estimated in this study is compared with precedential studies, as mentioned in L.129-133. As the local (national level) infrastructure risks are also quantified, could you show a comparison of some of these with previous studies which also analyze local risk?

We agree with the reviewer that a comparison on a national level would be very useful as well. Unfortunately, we cannot find any sufficiently similar national studies that show levels of risk on road or railway infrastructure asset damages. For example, Jongman et al (2012) shows road and

railway damages for a unique flood event in two places (Carlisle and Eilenburg), but it is near to impossible to compare this single flood event used in this study with the risk approach taken in our study.

We welcome any advice on studies that reviewers think are suitably comparable that we may have overlooked.

Jongman, B., Kreibich, H., Apel, H., Barredo, J. I., Bates, P. D., Feyen, L., ... & Ward, P. J. (2012). Comparative flood damage model assessment: towards a European approach. *Nat. Hazards Earth Syst. Sci*, 12, 3733-3752.

2. If my understanding is correct, the periodic maintenance and routine maintenance costs, maintenance period for each road, and the discount rate, are assumed without any relative references, which may present an additional uncertainty to the analysis. I think these assumptions also require sufficient reasoning, as is given for the other assumptions.

This a good point that we should have clarified. The assumptions on periodic and routine maintenance costs and period were taken from the calibration of the HDM4 model in various countries with recent World Bank investments (averaged to get just one set of assumptions). (ref: Kerali, Henry GR, Jennaro B. Odoki, and Eric E. Stannard. "Overview of HDM-4." *The Highway Development and Management Series 4* (2000).). The assumptions on discount rate were made to reflect different costs of capital in the various country groups, due to different growth rates and risk levels. They reflect the discount rates often used for public investments in these country groups. We have now added the following sentences in the methods section:

The upgrade costs are assumed to be 2% of the road value¹⁸, periodic maintenance is assumed to be occurring yearly with a cost of 0.075% of the road value, and the routine maintenance is assumed to happen every 6 years with a cost of 5% of the road value. These assumptions were taken from several calibrations of the HDM4 model⁵⁹ on various countries in which the World Bank recently invested in road projects.

3. Perhaps the impact of earthquakes is limited to soil liquefaction due to data limitations. I wonder if it would be possible, at least to consider the cost of removing trees fallen onto roads and railway tracks which were caused by earthquakes, by combining a global tree density map with the dataset related to earthquakes, along with that of cyclones. Otherwise, it should be clarified in L.56 that this study focuses solely on the risk of soil liquefaction associated with earthquakes to avoid any misunderstanding.

The reviewer is correct that the suggest damaged (in the sense of clean-up and repair costs) is not just limited to liquefaction, but also a myriad of other potential contributors such as trees falling over (as highlighted), structural damage, etc. In the case of this paper, we make the assumption that direct damages due to liquefaction (and related lateral spreading) is the major contributor of damage and clean-up costs following earthquakes to such widely spatially distributed transportation assets. In the instance where ground deformation is severe enough to topple a tree, we would also assume significant damages would similarly occur directly to the road/rail assets/corridors given the close proximity. The cost to remove a tree would be well absorbed within the repair cost of the road.

4. I would like the authors to provide further discussion regarding the relevance of the results with regard to current international policy schemes for alleviating the damage caused by natural disasters (e.g. SDGs 11.5, Sendai Framework for Disaster Risk Reduction 2015-2030 etc.) to reinforce the policy

implications of this study. How could the expected outcomes be useful for policy makers?

Thank you for this comment. We have added more global policy context in the introduction as well as a discussion on how these results can be useful to policy makers in the discussion section.

In addition, as minor comments,

1. L.257, 263: BCA -> BCR? Otherwise, please define it.

Yes, thank you. This is now changed to BCR.

2. L.282, 283: 45 per cent and 10 per cent may be uniformly written as 45% and 10%.

This is now changed.

3. L.283: Fig S7 -> Fig S6?

Thank you for picking this up. There was indeed a mistake between Fig S7 and Fig S6. We now name the CBA results Fig S6 (as it is referred earlier on in the text) and the results of the sensitivity analysis in Fig S7.

4. L.309: What is SREX? Please explain it.

SREX refers to the Special Report on Managing the Risks of Extreme Events and Disasters to Advance Climate Change Adaptation (SREX). We kept SREX in the text (because the report title is very long), but now added the reference.

5. L. 323: “~a changing climate.” some references should be added.

As references, we have added:

- Alfieri, L. et al. Global projections of river flood risk in a warmer world. *Earth’s Futur.* **5**, 171–182 (2017).
- Hirabayashi, Y. et al. Global flood risk under climate change. *Nat. Clim. Chang.* **3**, 816–821 (2013).
- Vousdoukas, M. I. et al. Global probabilistic projections of extreme sea levels show intensification of coastal flood hazard. *Nat. Commun.* **9**, 2360 (2018).

6. L.409: “Flood simulations are forced~(1/10 to 1/1000)” is redundant in the same paragraph.

We removed the duplicates in this paragraph. Apparently, something went wrong in merging comments and changes from different authors.

7. *It would be helpful for the reader to add a description for line and error bars in the box-and-whisker diagrams in Fig 3 B and C in the caption or figure.*

Good point! This is now added to both Figure 3 and 4 in the figure captions.

8. *Although I looked at Table S4, I am not sure of the difference between primary, secondary, and tertiary roads. Please clarify this within the main text.*

Agreed, we have now added an additional sentence in the first paragraph of “Global Infrastructure Data” to explain what we mean by primary, secondary and tertiary:

Primary roads can be described as all major highways and trunk roads. Secondary roads as all major provincial and subnational roads. Tertiary roads are considered to be important local roads, often linking secondary or primary roads with each other.

9. *For the caption of Table S7, I'm wondering if the table represents the design standards for not just pluvial flooding, but all flooding taken into account in this study.*

Yes the reviewer is right here. We have changed that now. For coastal and river flooding, we also consider the regions' overall flood protection standards, whereas for pluvial flooding we only assume the road's design standards.

10. *I think a world map representing which income class the regions belong to would be intuitive to complement Figs 1, 2, and 6.*

Good point! We have added this figure to the supplementary.

11. *Unfortunately, I could not understand how to estimate the sensitivity of parameters shown in Fig S6 even though I read the methodology section. In addition, "4L vs 2L" and "2L vs 1L" should be defined although I could surmise that they were related to the number of road lanes.*

We agree this could have been explained better. In the methods, we have now added lines below. In the caption of the figure in the supplementary materials, we have improved the explanation of the parameters.

For each asset type, a set of parameter values is determined using a Morris sampling approach⁵⁴, allowing for an optimal distribution of parameter values between the bounds. The results of the sensitivity analysis show how changes in each parameter that is included influences the estimate risk value.

Reviewer #3 (Remarks to the Author):

The manuscript presents a novel global analysis of the risks from multiple hazards affecting road and railway infrastructure, using recent and detailed global datasets and explores cost effective strategies for reducing risk. The overall analysis and outcomes, also on cost aspects and regional focus, are a useful and timely contribution that can be of interest to academics and practitioners across various disciplines. The authors are to be applauded for their work on the processing of the large and complex datasets required for the analysis. The methods are well described, including various aspects concerning the uncertainties, sensitivities and assumptions. The paper is clearly written. I believe the paper could be published after addressing the following points:

1. *General observation: After reading the manuscript, the reader has a good impression of the research, the assumptions and sensitivities in the used data, applied analysis and the results. However, after the first two paragraphs of the introduction, the reader is left in doubt about the importance of this research: is it mainly a minor (financial) issue for only a few countries or should the outcomes be considered a global alarming message with respect to foreseen future trends (more people, more infra, more traffic, more flooding, more extreme weather events?) and hence could improve planning, even though roads are not maintained globally, but tend to be managed by different levels of government within countries. So besides considering the research a starting point and aim for further research on the economic consequences, the authors could guide the reader by elaborating a bit more on the relevance of the results.*

We agree this could be explained better. We have rewritten the first three paragraphs of the introduction and the last paragraphs of the conclusion to improve the importance of the study and the relevance of the results

2. Line 29-32: To add to the relevancy of the work, I assume there should be more references available that relate to future infrastructure patterns and investment estimation (beyond 2025?), besides the preliminary paper included now?

See previous answer.

3. Line 57-61 / 416-429: The asset dataset is a crucial element in the analysis. The authors mention their source of infrastructure data is Openstreetmap, which, as far as I know, is a crowdsourced mapping initiative. Even though this dataset appears quite extensive, the following issues could be explained a bit more:

a. Do the authors know if there is a geographical bias in road coverage of the OSM dataset, potentially in relation to the crowdsourced nature of data gathering? Could there be consequences for the outcomes? The manuscript does not mention this. From Table S5 it appears that 40% of the roads used in the analysis are tertiary roads in high income countries (covering over 50% of all the roads used). Are the authors confident the road coverage in the other lower income groups (where hazard vulnerability is higher) is of the similar level/quality?

We agree that completeness of the data remains an issue when using open-source initiatives like OSM. As was identified by Barrington-Leigh et al., OSM was over 80% complete in January 2016, with missing data most likely in lower tier roads. Now, two years later, we assume completeness has substantially increased again. While working with the data, obvious caveats were the completeness of the lower tier roads in European countries (like the Netherlands or Germany, where closer to 100% of road surfaces are mapped) versus countries in Africa, where especially rural parts have missing roads. With this in mind, we excluded these lower tier roads from the global analysis. With regards to the amount of (tertiary) roads in high income countries versus low income countries: the main explanation for this difference is that there are also a lot more roads built in high income countries. Then again, most of the roads that are missing in less developed areas, will be unpaved roads. These roads are less likely to have high asset damages. Of course, flooding of these roads may have large impacts on local or regional communities. But as we focus purely on asset damages, we do not think our results will substantially change over time when more roads are mapped to OSM. Other unknown factors, such as the quality of the roads we consider, (paved vs unpaved) are much more important and are incorporated into our sensitivity analysis.

b. It would help the reader to assess the quality of the dataset if table S4 would include the km covered per category

We do the mapping in very early stages of the analysis. So it would be difficult to add kilometers per category in this table. In Table S5 you can find all the kilometers per aggregated category. Based on experience of working with this data, most OSM categories are either irrelevant to the study (such as *razed*, *dummy*, *raceway*, *proposed*, and *escape*), or are extremely small compared to the big categories (primary, secondary and tertiary).

c. The manuscript does not enclose any details on how the so-called planet-file was processed. All the code is now made available through GitHub <https://github.com/ElcoK/gmtra>, including in-line documentations. Additionally, a ReadTheDocs (<https://gmtra.readthedocs.io>) is available with further documentation.

4. Line 146-147: does the EAD per km also take into account differences between urban and rural roads, in the sense of traffic affected?

No, this study purely focuses on asset damages. We are currently working on expanding this work to include freight flows as well. Because there are no studies looking into global asset damages in such detail, we felt there was enough merit to publish this first. To avoid confusion, we have changed losses to damage throughout the entire manuscript.

5. Lines 174-174, Figure S1 and Table S1: According to Figure S1 and Table S1 the loss calculation and the assumptions are distinguishing between paved and unpaved roads. Can the authors explain how they applied this categorization to the OSM data? Same goes for the railway categorization mentioned in Table S1.

Thank you for picking this up. It seems we forgot to explain this in the text. We have now added the following lines to the Methods section (see subheading Roads and railways, under Estimation of fragility, damage and risk) to explain this:

Unfortunately, for most of the roads extracted from OSM, we do not know whether they are paved or unpaved. By using the kilometers of paved and unpaved roads from the CIA World Factbook, we were able to get the percentage of paved roads for each country. This, however, only gave us a total percentage based on all roads in the country. To get a percentage for each road type in a country, we compared the share of each road type with this total percentage of paved roads. Let us explain this through an example. Let's say the total percentage of paved roads in a country is 80%. We first compare this number with the percentage of primary roads in the country. If the percentage of primary roads in a country is 30%, we assume that all the primary roads are paved. This means we have 50% of paved roads 'left'. Now we compare this with the secondary roads. Let's say 30% of the roads in the country are secondary roads. Following the same reasoning, this means all the secondary roads are paved and we have 20% of paved roads 'left'. The share of tertiary roads in this country is 40%. As we only have 20% of paved roads 'left', we know assume that half of the tertiary roads are paved and the other half is unpaved. All other roads in the country are considered to be unpaved. For many countries, it is unknown what the percentage of electrified railway is in that country. As such, the percentage electrified vs non-electrified is fully incorporated into the sensitivity analysis.

6. Line 167 and Table S7: are tertiary roads considered part of "other"? This seems confusing with the "other roads" category in Table S5.

Good point, we now explicitly mention tertiary in Table S7.

7. Line 77 and 337: Besides a technical reason, the authors do not further explain why they have done the analysis at the level of the GADM regions and not, for instance, report the outcomes on a raster with equal cell sizes. How was this aggregation done, this is not explained? The regions vary in size, shape and characteristics, which makes the maps in Figures 1, 2 and 6 a bit challenging to interpret, while most results are also presented on a country or WB region level.

This is indeed a practical/technical reason, but we also decided to stick with administrative areas as policy makers tend to look at administrative areas, defined regions. This way, the results are easily interpreted for a certain region, group of regions/country. A reader is freely open to manipulate the input regions provided code (<https://github.com/ElcoK/gmtra>) to a gridded surface should they have a particular area of interest which would benefit from an equal-area raster map.

8. *Figure 5: How is the value of the country infrastructure, as displayed in Figure 5c, determined? I could not find a description for that.*

Thank you for picking this up. To estimate the infrastructure value, we used a similar approach as estimating the possible damage to a road. For the infrastructure value, we would also need to know the type, the width and whether it's paved or unpaved. As such, we again take the set of parameters we use for the sensitivity analysis, and use this to calculate a range of possible infrastructure values for each road asset. To get one value, we taken the median of these outcomes. As there are no readily available statistics on infrastructure values per country, and to make sure we have values that are consistent along all countries, we believe this is the best we could do.

Minor remarks:

9. *Lines 197-198: duplication of text*

Thank you, duplicate line of text is removed.

10. *Line 164: I think this points to the wrong figure?*

Thank you, you are right. It should be Fig S5. We shifted some figures around and we forgot to update this one.

11. *Line 257 and 263: BCA? Do you mean BCR?*

Yes, thank you. This is now changed to BCR.

12. *Line 283: I think this points to the wrong figure?*

You are right, this is now fixed.

13. *Line 301: 60 mln km? I thought "other roads" were not included (line 426-427), so shouldn't this say ~50 mln km.*

You are right, this is now changed.

14. *Lines 405-409 / 409-415: duplication of text*

Duplicates have been removed.

REVIEWERS' COMMENTS:

Reviewer #2 (Remarks to the Author):

All of my comments have been addressed adequately, and I think the current manuscript is acceptable for publication.

Reviewer #3 (Remarks to the Author):

It was interesting to read the revised manuscript and take note of the updates. The authors have adequately addressed the comments and, where relevant, sufficiently added information to the manuscript to increase importance of the work and to provide clarification of several issues raised in the comments.

Overall response to reviewers

Reviewer #2 (Remarks to the Author):

All of my comments have been addressed adequately, and I think the current manuscript is acceptable for publication.

We are happy to see that the reviewer acknowledges the work we put in the revisions

Reviewer #3 (Remarks to the Author):

It was interesting to read the revised manuscript and take note of the updates. The authors have adequately addressed the comments and, where relevant, sufficiently added information to the manuscript to increase importance of the work and to provide clarification of several issues raised in the comments.

We are glad the reviewer is happy with the changes we made to the manuscript based on her/his careful review of the previous version of the manuscript.